# Navigating the Deployment Dilemma and Innovation Paradox: Open-Source v.s. Closed-source Models

## Abstract

Recent advances in Artificial Intelligence (AI) have introduced a new paradigm in Machine Learning (ML) model development: pre-training of foundation model and domain adaptation. Two groups lead in developing foundation model: closed-source developers and open-source community. As open-source community becomes increasingly engaged, the performance open-source models are catching up with closed-source models. However, this leaves domain deployers into a dilemma: use closed-source models via API access or host open-source models on proprietary hardware. Using closed-source models incurs recurring costs, while hosting open-source models incurs substantial hardware investments and potentially lagging advancements. This paper presents a game-theoretical model to examine the economic incentives behind the deployment choice and the impact of open-source engagement strategy on technology innovation. We find that the deployer consistently opts for closed-source APIs when the open-source community engages in the market reactively by maintaining a fixed performance ratio relative to closed-source advancements. However, open-source models can be favored when a proactive open-source community produces high-performance models independently. Also, we identify conditions under which engagement and competitiveness of the open-source community can foster or inhibit technological progress. These insights offer valuable implications for market regulation and the future of AI model innovation.

## CCS Concepts

• **Computing methodologies**;

## Keywords

Deployment dilemma; open-source; closed-source; foundation model

**ACM Reference Format:**
Anonymous Author(s). 2018. Navigating the Deployment Dilemma and Innovation Paradox: Open-Source v.s. Closed-source Models . In *Proceedings of Make sure to enter the correct conference title from your rights confirmation emai (Conference acronym 'XX)*. ACM, New York, NY, USA, 14 pages. https://doi.org/XXXXXXX.XXXXXXX

## 1 Introduction

The capability of general AI, especially Large Language Models (LLMs) has seen a remarkable surge due to scaling of training data,

compute, and model parameters [23, 40]. Most recently, the paradigm of pre-training and domain adaptation has become increasingly important in LLM development [25? ? ]. As the landscape of foundational models is characterized by two prominent alternatives: open-source and closed-source. The domain expert would always make decision about which technology to adopt. Thus, a development of end technology always follow the process of pre-training, deployment, and adaptation.

The foundational model market is increasingly competitive primarily due to the emergence of open-source models. Take LLMs as an example. Stanford has reported that of the 149 foundation models released in 2023, 98 were open-source models such as LLaMA [39] and 23 were closed-source with a public API to access such as GPT-4[30, 31]. Importantly, there has been a significant increase in the proportion of models released with open access [30]. Clearly, the engagement of open-source community form up a competitive landscape for the foundational model development[1, 9].

The relationship between a competitive market and innovation is complex, with competition capable of both stifling and fostering innovation [15, 38]. Notably, competition between open-source and closed-source models presents unique dynamics distinct from typical firm-to-firm competition. Unlike traditional corporate players, open-source communities often operate with diverse motivations beyond profit, such as community-driven improvement, accessibility, and transparency [4, 18, 29, 35]. This makes the impact of open-source versus closed-source competition on technological innovation particularly intricate. Understanding how these competing models influence the trajectory of technological progress is essential, as it can reveal insights into the forces that drive or inhibit advancements within foundation models, with implications for future policy and innovation strategy.

Besides, due to the engagement of open-source community, deployers would face a deployment dilemma, navigating complex economic trade-offs in choosing which technology to adopt. In one aspect, self-hosting open-source technology comes with the high cost associated with the requisite hardware resource such as GPUs while using third-party API leads to recurring cost [13]. In addition, the performance of the foundation technology directly impacts that of the end technology, which, in turn, affects the revenue generated in the end market [25]. Thus, the choice between self-hosting open-source technology or utilizing third-party APIs involves a complex trade-off from an economic perspective. Understanding this process is necessary to study the economic and technological consequences of the open-source technology.

In this paper, we present a comprehensive game-theoretic model to explore the interactions among closed-source developers, open-source communities, and deployers and how these interactions affect the competitive and innovative outcomes of foundation model

development. We analyze three distinct scenarios: a baseline scenario **without open-source engagement**, a scenario with **proactive open-source engagement** where the community independently innovates and decides the model performance, and a scenario with **reactive open-source engagement** in which the open-source community aligns its performance to maintain relative parity with closed-source advancements.

Our analysis shows that deployment choice always falls into one of three primary outcomes: **API-dominant**, where open-source engagement has no impact on the market status or decisions of closed-source developers compared to scenarios without an open-source alternative; **API-strategic**, where open-source engagement prompts strategic behaviors from closed-source developers, yet deployers are still incentivized to adopt closed-source technology; and **self-hosting**, where open-source technology fully supersedes the closed-source option. It turns out that the outcomes are highly dependent on the open-source engagement strategy. Our findings indicate that open-source engagement can significantly alter the innovation landscape for foundation models. In particular, we identify conditions where open-source competition paradoxically hampers innovation by discouraging closed-source developers from pushing foundational advancements, as well as cases where it promotes a "race-to-the-top," encouraging closed-source developers to innovate aggressively.

The main contributions of this paper are threefold. First, we provide a theoretical framework based on **multi-stage game** and **subgame perfect equilibrium** to analyze the deployment dilemma facing model deployers and define three types of deployment outcomes. Second, we investigate two distinct open-source engagement strategies - proactive or reactive - and characterize the conditions under which the engagement would encourage or inhibit innovation in foundational technologies. Finally, we discuss broader implications for policymakers, offering insights into how the regulation of open-source and closed-source model competition can support sustainable AI innovation.

## 2 Related Work

There exists extensive research on technology innovation and competition. Our work specifically examines the dynamics of competition and innovation between open-source and closed-source models within the paradigm of pre-training and fine-tuning. Taking an economic perspective, we are the first to explore how open-source technology drives competition and impacts the trajectory of technological innovation.

**Open-Source Community.** Open-source community has led to great technological advances and unprecedented global collaboration by providing open-source software (OSS), to which everyone can have free access [5, 12]. In the last decade, open-source community has been a driven force of the development of artificial intelligence. For example, 98 of the 149 foundation LLMs released in 2023 were open-source models such as LLaMA [30, 39]. Moreover, recent researches have shown that open-source LLMs is quickly catching up closed-source commercial LLMs and the performance gap can be supplemented or even closed with appropriate adaptation techniques, such as adapting [1, 9]. Importantly, the incentives of open-source have been discussed and proven to be far more

beyond profit[4, 18, 29, 35]. Despite the significant role of OSS, there remains a scarcity of research that quantitatively assesses its value [19]. Our work also contributes to the literature by providing insights how open-source community's engagement in the market and its innovation strategies influence the market dynamics and technology outcomes.

**Technology Deployment.** For domain-specific deployers, it has been tricky to make a decision between self-hosting open-source technology or closed-source technology. Adopting API may lead to concerns such as data ownership, privacy and stability [9, 10]. However, self-hosting can be extremely expensive due to high hardware requirements. For example, "regular 16-bit adapting of a LLaMA 65B parameter model requires more than 780 GB of GPU memory" [13]. Moreover, the model performance has crucial influence on the adoption decision. For example, open-source options can incur extra adaptation costs as their foundational capability lags behind, while cloud APIs also limit the developer's ability to adapt the models with custom data [21]. Considering the complexity of adoption decision, our paper is the first to model the deployment dilemma from an economic perspective.

**Economic Models of Technology Innovation**. Many works addressed the technological innovation, production, and cooperation between firms. Empirical studies have proven that the relationship between competition can either have a negative or positive relationship with innovation, depending on multiple factors such as market structure and innovation strategis[15, 38]. However, the incentives of innovation are restricted to profit difference a firm can earn with more innovation compared to with less innovation, which is not applicable for open-source community. Bhaskaran and Krishnan [3] provide a model of joint work and decision making between collaborating firms for new product development. However, it focuses on cooperation rather than competition between firms, and the innovation process does not follow the pre-training and adaptation paradigm.

**Machine Learning and Game Theory**. Our paper generally contributes to the work that uses game theory to analyze the economics of technology, especially ML models [24, 28, 32]. Specifically, our work contributes to the study of technology competition and innovation with open-source engagement. Kleinberg et al. [25] propose a model of fine-tuning of general-purpose technology. However, it focuses on joint development and bargaining of two firms and ignores the open-source engagement and competition.

## 3 Model

To capture the interactions between players, particularly the role of the open-source community's engagement and strategies, we propose three game-theoretic models: a baseline model and two variations. The baseline model includes only a closed-source foundation model developer, a domain-specific deployer, and end users. In each variation model, an open-source community is introduced, each employing distinct strategies in technology innovation. Both the closed-source developer and the open-source community focus on advancing general-purpose technology, while the deployer must select one technology for deployment.

## 3.1 Model Setting

In this section we introduce the agents and the assumptions associated with each of them in the model.

**Closed-source Technology Developer**. The developer $F$ develops a closed-source foundation model to performance level $\alpha_0 \in \mathbb{R}_0^+$ and prices the API as $\gamma c$ ($\gamma > 1$) by deciding a multiplier $\gamma$, where $c \in \mathbb{R}^+$ is the unit cost of operation.

**Open-Source Community**. The open-source community $O$ provide a technology at level $\alpha \in \mathbb{R}_0^+$ for free adoption. As the incentives surrounding open-source technology can be complex, we define its engagement strategy as *reactive* or *proactive* strategy. Specifically, reactive engagement means $O$ always follows the closed-source technology and maintains the relative performance of open-source technology to closed-source technology, according to a parameter $m$, thus $\alpha = m\alpha_0$. In contrasts, proactive engagement means $O$ always decides the performance level of open-source model $\alpha$ independent of $\alpha_0$.

**Domain-specific Deployer**. The deployer $S$ first decides whether to host the open-source technology or use the API, captured by a variable $I$: when $I = 1$, the deployer chooses the closed-source API; when $I = 0$, the deployer opts for the open-source option. Then, it adapts the technology to level $\alpha_1 \in \mathbb{R}_0^+$ where $\alpha_1 \geq \alpha_0$ and set the unit price of the technology as $p \in \mathbb{R}_0^+$. Notably, $S$ would have to operate on its own hardware infrastructure if host open-source technology and buy the API in demand if use the closed-source technology.

**End Users**. The end users $U$ reacts to the technology with demand $D$. Drawn from literature investigating customer consumption behavior, the market demand is determined by the technology's price $p$ and technology's performance $\alpha_1$ at the same time [17, 36, 41]. We assume there is a function $D : \mathbb{R}_0^+ \times \mathbb{R}_0^+ \to \mathbb{R}^+$ such that $D(p, \alpha_1)$ is the demand in the end market with technology at level $\alpha_1$ and unit price $p$. $D(p, \alpha_1)$ is monotonically increasing with $\alpha_1$ and monotonically decreasing with $p$. We assume the demand function is publicly known.

**Revenue**. Revenue is calculated as demand multiplied by unit price [34]. The developer $F$ gains a revenue $R_F = \gamma c D(p, \alpha_1)$ by providing inference API to $S$. The deployer $S$ gains a revenue $R_S = p D(p, \alpha_1)$ from the end market.

**Cost**. Both $F$ and $S$ have two parts of cost: technology production (development or adaptation) cost and operation cost. $F$ has a development cost $C_F(\alpha_0) : \mathbb{R}_0^+ \to \mathbb{R}_0^+$ to produce a general technology at level $\alpha_0$ and an operation cost of $c$ per unit. $S$ faces a adapting cost $C_S^{api}(\alpha_1; \alpha_0) : \mathbb{R}^+ \to \mathbb{R}^+$ to adapt the closed-source technology from level $\alpha_0$ to $\alpha_1$ or a cost function $C_S^{self}(\alpha_1; \alpha) : \mathbb{R}_0^+ \to \mathbb{R}_0^+$ to adapt the open-source technology from level $\alpha$ to $\alpha_1$. Besides, $S$ faces an operation cost of $c$ per unit if self-hosts open-source technology or $\gamma c$ per unit if uses API. We assume these cost functions are publicly known.

**Utility**. The utility of developer $F$, denoted as $U_F$, and of deployer $S$, denoted as $U_S$, are calculated by $(R_i - C_i)$, where $i = S, F$.

$$U_F^{api} = (\gamma c - c)D(p, \alpha_1) - C_F(\alpha_0) \tag{1}$$

$$U_S^{api} = (p - \gamma c)D(p, \alpha_1) - C_S^{api}(\alpha_1; \alpha_0) \tag{2}$$

$$U_S^{self} = (p - c)D(p, \alpha_1) - C_S^{self}(\alpha_1; \alpha) \tag{3}$$

$$U_F^{self} = 0 \tag{4}$$

We introduce the following notations for utility: $U_S^{api}, U_S^{self}, U_F^{api}, U_F^{self}$ to represent the utilities of the deployer $S$, the developer $F$ under both the API and self-hosting scenarios.

**Technology Innovation Outcome (Society Level)**. At the societal level, the technology innovation outcome is defined as

$$\alpha_0^{soc} = I\alpha_0 + (1 - I)\alpha \tag{5}$$

$$\alpha_1^{soc} = \alpha_1. \tag{6}$$

## 3.2 Game Process

The game process varies according to the open-source community's engagement and strategic choices, resulting in three distinct models: the baseline model without open-source community involvement, and two variations where the community adopts proactive or reactive strategies. These models are summarized below and illustrated in Figure 1.

**Baseline Game - No Open-Source Engagement**. Here, the open-source community $O$ chooses not to engage in the market. Thus, developer $F$ first brings the foundation technology to performance level $\alpha_0$ and sets the unit price for API usage as $\gamma c$ by deciding the multiplier $\gamma$. Then, $S$ adapts the technology to level $\alpha_1$ and sets the end-user price $p$. The end users consume the technology with demand $D(p, \alpha_1)$. Revenue is generated for both $F$ (through API usage fees) and $S$ (through end-user sales), highlighting the dynamics of a market without competition.

**Game 1 - Proactive Open-Source Engagement**. Here, the open-source community $O$ adopts a proactive engagement strategy. First, $O$ independently develops its technology to reach a performance level $\alpha$. Second, the closed-source developer $F$ establishes its own foundation technology at level $\alpha_0$ and sets the API unit price as $\gamma c$ by choosing the multiplier $\gamma$. The deployer $S$ then chooses between self-hosting the open-source technology or accessing the closed-source technology via API. After deployment, $S$ adapts the selected technology to a domain-specific level $\alpha_1$ and sets the end-user price $p$, resulting in demand $D(p, \alpha_1)$ from end users. The consumption of end technology generates revenue for the deployer, and for the developer as well, but only if the deployer opts for the closed-source technology.

**Game 2 - Reactive Open-Source Engagement**. Here, the open-source community $O$ follows a reactive engagement strategy. It initially announces this approach by specifying a performance ratio $m$ to indicate how closely it will track the closed-source technology developed by $F$. Once $F$ has finalized its technology at level $\alpha_0$ and sets the API price as $\gamma c$, $O$ develops its technology to level $m\alpha_0$. The subsequent deployment, adaptation, and consumption steps are identical to those in Game 1.

## 3.3 Solution of the Model

In this section, we provide the general equilibrium of each model derived through backward induction, following the sequential decision-making of the deployer $S$ and the developer $F$. The solution involves two key steps.

**Step 1:** Assuming a fixed $\alpha_0$ and $\gamma$ (or also $\alpha$), $S$ maximizes its utility by choosing the optimal domain technology performance

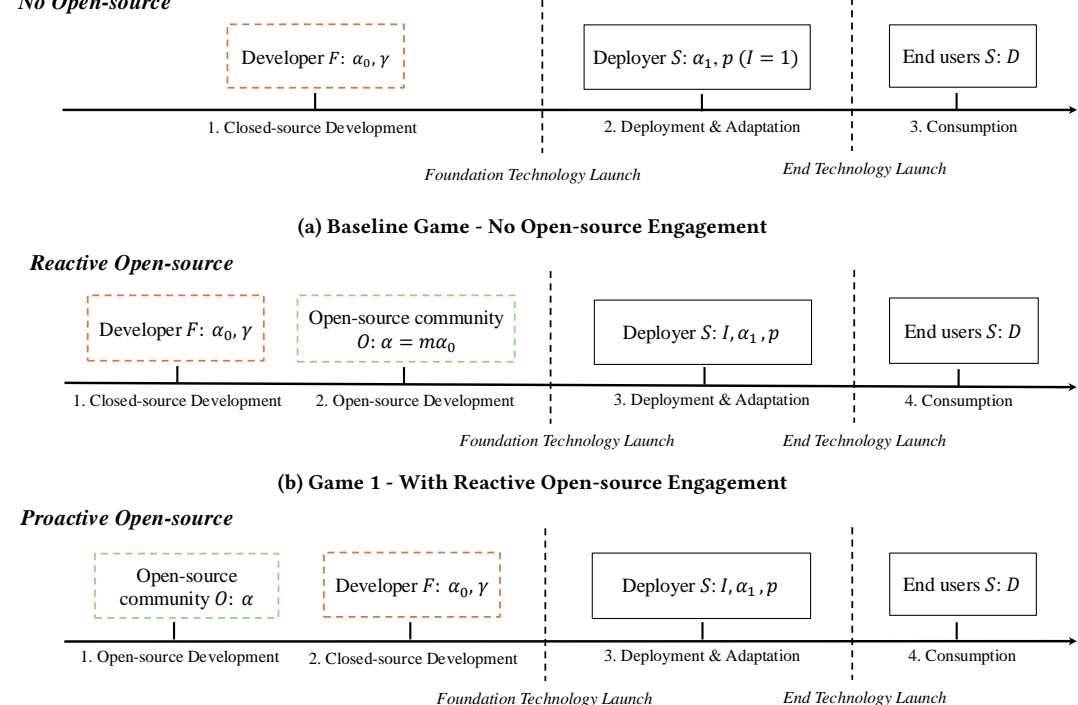

**Figure 1: An illustration of the processes for the three games. Game 1 and Game 2 differ from the baseline model in the foundation technology development stage, as they involve open-source community. In Step 2 of the baseline game, $I = 1$ always holds, whereas in Game 1 and Game 2, $I$ can be either 0 or 1, reflecting the deployer's deployment decision. The distinction between Game 1 and Game 2 arises from the strategy adopted by the open-source community.**

level $\alpha_1$ and price $p$. Formally, $S$ solves the following optimization problem:

$$I^*, \alpha_1^*, p^* = \arg\max_{I, \alpha_1, p} I U_S^{api} + (1 - I) U_S^{self}. \tag{7}$$

The deployer $S$ will choose to self-host the technology if its utility from self-hosting, denoted as $U_S^{self}(\alpha_1^*, p^*)$, is greater than its utility from using the API service, denoted as $U_S^{api}(\alpha_1^*, p^*)$.

**Step 2 - Baseline Game:** Anticipating $S$'s response to its decisions regarding the foundational performance level $\alpha_0$ and the inference price parameter $\gamma$, $F$ sets $\alpha_0$ and $\gamma$ to maximize its own utility. This leads to the following optimization problem for $F$:

$$\alpha_0^*, \gamma^* = \arg\max_{\alpha_0, \gamma} U_F^{api}(\alpha_1^*, p^*). \tag{8}$$

**Step 2 - Game 1 and Game 2:** Similar to the step 2 in the baseline game, $F$ would optimize its utitlity by deciding:

$$\alpha_0^*, \gamma^* = \arg\max_{\alpha_0, \gamma} U_F^{api}(\alpha_1^*, p^*). \tag{9}$$

Since the developer $F$ receives revenue only if $S$ decides to use the API, Step 2 is only meaningful when I = 1, which means

$$U_S^{self}(\alpha_1^*, p^*) \geq U_S^{api}(\alpha_1^*, p^*). \tag{10}$$

Also, $F$ would participate in the development only when it expect to gain a positive utility, as $U_F^{api}(\alpha_0^*, \gamma^*, \alpha_1^*, p^*) \geq 0$. Else, $F$ would anticipate no opportunity to gain a profit and exit the market.

The solution depends on market conditions and the engagement strategy of $O$. Thus, we offer a set of relevant definitions to help characterize the different possible regimes of solutions according to the developer's strategic behavior and deployer's deployment decisions.

*Definition 3.1 (API-DOMINANT SOLUTION).* The **API-dominant solution** is the solution when developer $F$'s optimal decision, $\alpha_0^*, \gamma^* = \arg\max_{\alpha_0, \gamma} U_F^{api}$, naturally satisfies $U_S^{self} \geq U_S^{api}$ and $U_F^{api} \geq 0$. In this situation, $S$ chooses API choice as it naturally dominates the self-hosting choice.

*Definition 3.2 (API-STRATEGIC SOLUTION).* The **API-strategic solution** is the solution when developer $F$'s optimal decision, $\alpha_0^*, \gamma^* = \arg\max_{\alpha_0, \gamma} U_{F, api}$ under the constraint $U_S^{self} \geq U_S^{api}$ naturally satisfies $U_F^{api} \geq 0$, while only $\alpha_0^*, \gamma^* = \arg\max_{\alpha_0, \gamma} U_{F, api}$ leads to $U_S^{self} < U_S^{api}$. In this situation, $F$ strategically incentivizes $S$ to choose the API by ensuring that $S$ achieves greater profit through the API option compared to the self-hosting alternative.

*Definition 3.3 (SELF-HOSTING SOLUTION).* The **self-hosting solution** is the solution when no combination of $\alpha_0^*, \gamma^*$ exists that simultaneously satisfies: $U_S^{self} \geq U_S^{api}$ and $U_F^{api} \geq 0$ . In this situation, $S$ opts to self-host the open-source technology rather than utilize the API provided by $F$.

Notice that any solution will fall into one of three categories: an API-dominant solution, an API-strategic solution, or a self-hosting solution. These regimes are shaped by the market conditions and the engagement strategy of $O$. Analyzing the general form is challenging due to multiple sequential decision steps, each requiring the consideration of multiple factors. At each stage, either the developer or deployer must determine optimal values for variables such as performance levels, pricing, and deployment choices, which interact in complex ways across stages. This interdependence makes deriving a general solution intricate, necessitating specifications to gain clearer insights. Accordingly, in the next section, we present formal theorems under specified demand and cost functions.

# 4 Analysis of Separable Multiplicative Demand and Quadratic Cost

In order to produce closed-form solutions and understand how the agents in the model interact with each other, we take the form of separable multiplicative demand and quadratic cost, which are commonly used in business research.

The demand function is expressed in the separable multiplicative form: $D(p, \alpha_1) = d_1(p) * d_2(\alpha_1)$, where $d_1(p)$ measures the effect of price and $d_2(\alpha_1)$ represents the effect of quality [2, 11]. For the price-dependent part, linear model has been extensively used in economics and business literature, including theoretical models [6, 14, 33, 34, 37] and empirical estimations [7, 20]. For the quality effect, we take the form as $d_2(\alpha_1) = \alpha_1$ [27]. Thus, we get

$$D(p, \alpha_1) = (a - bp)\alpha_1 \quad (11)$$

, where $a > 0$ and $b > 0$ are constant parameters representing the market size and price sensitivity respectively. The demand would decrease with price and increase with product quality. Also, there would be no sales for zero quality (performance level).

The quadratic form for modeling cost is widely adopted in economics and management science literature [3, 8, 16, 22, 26, 41]. Following Kleinberg et al. [25], we assume that the cost increases quadratically with advancements in technology:

$$\phi(\alpha_0) = K_F \alpha_0^2 \quad (12)$$

$$\phi(\alpha_1; \alpha_0) = K_S^{api}(\alpha_1 - \alpha_0)^2 \quad (13)$$

$$\phi(\alpha_1; \alpha) = K_S^{self}(\alpha_1 - \alpha)^2 \quad (14)$$

Here, $K_F$, $K_S^{self}$, and $K_S^{api}$ are positive constants, reflecting that marginal costs should increase with technology advancement [25]. As $K_F$ and $K_S^{self}$ include both non-hardware and hardware costs while $K_{S,api}$ includes only non-hardware costs. The cost factors are decomposed as:

$$K_F = K_{PRE} + K_G \quad (15)$$

$$K_S^{api} = K_{FT} \quad (16)$$

$$K_S^{self} = K_{FT} + K_G \quad (17)$$

- $K_{PRE}$ represents the non-hardware cost component in the pre-training cost $K_F$
- $K_{FT}$ represents the non-hardware cost component in the adapting cost $K_S^{self}$ and $K_S^{api}$
- $K_G$ represents the hardware cost component in $K_F$ and $K_S^{self}$

Thus, the utilities of developer $F$, deployer $S$, and end user $U$ are as

$$U_S^{api} = (p - \gamma c)(a - bp)\alpha_1 - K_S^{api}(\alpha_1 - \alpha_0)^2 \quad (18)$$

$$U_F^{api} = (\gamma c - c)(a - bp)\alpha_1 - K_F \alpha_0^2 \quad (19)$$

$$U_S^{self} = (p - c)(a - bp)\alpha_1 - K_S^{self}(\alpha_1 - \alpha)^2 \quad (20)$$

$$U_F^{self} = 0 \quad (21)$$

## 4.1 Equilibrium without $O$

THEOREM 4.1 (API-DOMINANT STRATEGY). *Without $O$, the equilibrium always falls into the API-dominant solution, yielding the following strategies:*

$$\gamma^* = \frac{5\theta + 3 - 2\beta(3 + \theta) - \sqrt{\delta}}{8(1 - \beta)}, \quad (22)$$

$$\alpha_0^* = \frac{b}{4K_F}\left(c\gamma^* - c\right)\left(\frac{a}{b} - c\gamma^*\right), \quad (23)$$

$$p^* = \frac{a}{b} + c\gamma^*, \quad (24)$$

$$\alpha_1^* = \alpha_0^* + \frac{b\left(\frac{a}{b} - c\gamma^*\right)^2}{8K_{FT}}, \quad (25)$$

*where:*

$$\delta = (5\theta + 3 - 2\beta(3 + \theta))^2 - 16(1 - \beta)(\theta^2 + 3\theta - 2\beta(1 + \theta)),$$

$$\theta = \frac{a}{bc}, \quad \beta = \frac{K_{FT}}{K_{PRE} + K_G}.$$

A proof of the above result is provided in Appendix A.1. Notice that the deployer $S$'s decision on domain-specific technology performance $\alpha_1^*$ equals $\alpha_0^*$ plus $\frac{b\left(\frac{a}{b} - c\gamma^*\right)^2}{8K_{FT}}$, which is independent of developer $F$'s decision on foundation technology performance $\alpha_0^*$. This finding is consistent with the finding from a previous research by Kleinberg et al. [25]. Moreover, both the developer $F$'s decision on foundation technology performance $\alpha_0^*$ and deployer $S$'s decision on domain-specific technology performance $\alpha_1^*$ are independent of the open-source technology performance, which is reasonable as the open-source technology is naturally dominated by the closed-source technology and cannot influence the market. The results are shown in Appendix Figure B.1.

## 4.2 Subgame Perfect Equilibrium with a Reactive $O$ under a Fixed $m$

When the open-source community adopts a reactive strategy, the subgame perfect equilibrium under a given $m$ may lead to different solutions based on various market factors, captured by cost parameters $\{K_G, K_{FT}, K_{PRE}, c\}$ and market consumption parameters $\{a, b\}$. Among all these factors, we focus on $K_G$, which indicates the hardware cost. First, we specify the forms of solutions. Then, we show how $K_G$ and $m$ characterize the equilibrium solution with a focus on technology outcome.

**Theorem 4.2.** *With a reactive O, the API-dominant solution results in strategies that are identical in form to those presented in Theorem 4.1.*

This conclusion is straightforward, as under an API-dominant solution, the engagement of the open-source community does not affect the dynamics of the original game, leaving the strategic outcome unchanged.

**Theorem 4.3** (API-Strategic Solution with reactive O). *With reactive O, the API-strategic solution yields strategies as follows:*

$$I^* = 1,$$

$$p^* = \frac{1}{2}\left(\frac{a}{b} + c\gamma^*\right),$$

$$\alpha_1^* = \alpha_0^* + \frac{b\left(\frac{a}{b} - c\gamma^*\right)^2}{8K_{FT}},$$

*and $\alpha_0^*$ and $\gamma^*$ is the solution of:*

$$\begin{cases} \left(16(\theta - \gamma^*)^2 - m(\theta - 1)^2\right)\alpha_0^* = \left(\frac{(\theta-1)^4}{K_{FT}+K_G} - \frac{(\theta-\gamma^*)^4}{K_{FT}}\right)bc^2, \\ \frac{2\left(bc^2(\gamma^*-1)(\theta-\gamma^*) - 4(K_{PRE}+K_G)\alpha_0^*\right)}{\alpha_0^*(\theta+1-2\gamma^*) + \frac{bc^2}{8K_{FT}}(\theta-\gamma^*)^2(3+\theta-4\gamma^*)} = \frac{-bc^2\left((\theta-\gamma^*)^2 - m(\theta-1)^2\right)}{\frac{bc^2}{8K_{FT}}(\theta-\gamma^*)^3 + (\theta-\gamma)\alpha_0^*}, \end{cases}$$

*, where $\theta = \frac{a}{bc}$.*

A proof of Theorem 4.3 is provided in Appendix A.3. Note that the existence of a feasible solution in Theorem 4.3 is guaranteed by Theorem **??**, while its uniqueness is ensured by the maximization of $U_F$.

**Theorem 4.4** (Guaranteed API outcome). *With reactive O, the equilibrium always falls into either a API-dominant or a API-strategic solution, meaning there always exists a combination $\{\alpha_0^*, \gamma^*, \alpha_1^*, p^*\}$ that satisfies $U_S^{self} \geq U_S^{api}$ and $U_F \geq 0$ simultaneously.*

A proof of Theorem 4.4 is provided in Appendix A.5. Notably, when $O$ adopts a reactive strategy, $F$ can influence technology innovation in a way that strategically deters $O$ and encourages $S$ to adopt the closed-source technology. Counterintuitively, even when $m$ is high—indicating that the open-source technology significantly outperforms the closed-source technology—the deployer $S$ is still incentivized to utilize the closed-source technology via API.

Next, we illustrate the impact of reactive open-source engagement on technology outcomes using numerical results. We set parameters of ($a = 8, b = 1, c = 0.5, K_{FT} = K_{PRE} = 1$) and let $m$ range from 0.1 to 1.4.

As shown in Figure 2a, foundation technology innovation is generally hindered when $m$ is high. This is because, at higher $m$ values, the closed-source developer may choose to strategically reduce technology performance to deter open-source alternatives. When $m$ decreases to a low level, the closed-source developer can gain higher technology advantage by enhancing performance, which is an economical strategy to attract deployers toward the closed-source API.

Interestingly, we observe in Figure 2b that end technology experiences higher levels of innovation. This outcome arises because the closed-source developer not only adjusts technology performance but also lowers the API price, allowing the deployer to achieve

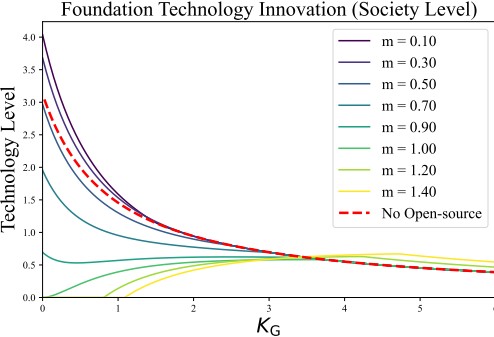

(a) **Foundation Technology**

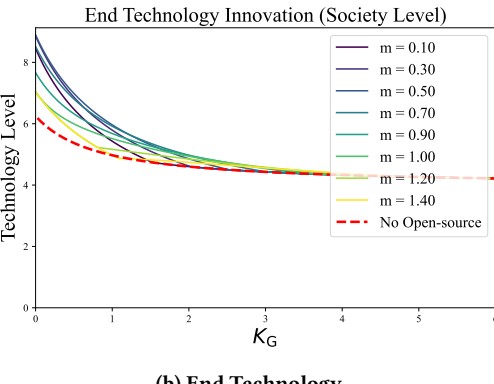

(b) **End Technology**

**Figure 2: Technology Outcomes Comparison - No Open-source vs. Reactive Open-source Engagement ($a = 8, b = 1, c = 0.5, K_{FT} = K_{PRE} = 1$)**

higher unit profit from end technology. This incentivizes the deployer to further adapt the technology to an enhanced level, thus driving end technology innovation.

### 4.3 Subgame Perfect Equilibrium with a Proactive $O$ under a Fixed $\alpha$

Similar to the previous section, we first specify the forms of each solution under a fixed $\alpha$ and then analyze how $K_G$ and $\alpha$ characterize the equilibrium solution.

**Theorem 4.5.** *With a proactive O, the API-dominant solution results in strategies that are identical in form to those presented in Theorem 4.1.*

**Theorem 4.6** (API-Strategic Solution with proactive O). *With proactive O, the API-strategic solution yields strategies as follows:*

$$I^* = 1,$$

$$p^* = \frac{1}{2}\left(\frac{a}{b} + c\gamma^*\right),$$

$$\alpha_1^* = \alpha_0^* + \frac{b\left(\frac{a}{b} - c\gamma^*\right)^2}{8K_{FT}},$$

*and $\alpha_0^*$ and $\gamma^*$ is the solution of:*

$$\begin{cases} 16\left((\theta-\gamma^*)^2\alpha_0^* - (\theta-1)^2\alpha\right) = \left(\dfrac{(\theta-1)^4}{(K_{FT}+K_G)} - \dfrac{(\theta-\gamma^*)^4}{K_{FT}}\right)bc^2, \\[2ex] \dfrac{bc^2(\gamma^*-1)(\theta-\gamma^*)-4(K_{PRE}+K_G)\alpha_0^*}{\alpha_0^*(\theta+1-2\gamma^*)+\frac{bc^2}{8K_{FT}}(\theta-\gamma^*)^2(3+\theta-4\gamma^*)} = \dfrac{-4bc^2(\theta-\gamma^*)}{8\alpha_0^*+\frac{bc^2}{K_{FT}}(\theta-\gamma^*)^2}, \end{cases}$$

*,where $\theta = \dfrac{a}{bc}$.*

A proof of Theorem 4.6 is provided in Appendix A.4. Note that the solution from Theorem 4.6 must always satisfy $\alpha_0^* \geq 0$, $\gamma^* \geq 1$, and $U_F(p^*, \alpha_1^*, \alpha_0^*, \gamma^*) \geq 0$. If these conditions are not met, the equilibrium defaults to the self-hosting solution described below.

THEOREM 4.7 (SELF-HOSTING SOLUTION WITH PROACTIVE O). *With a proactive O, a self-hosting solution yields the following strategies:*

$$\gamma^* = 1,$$
$$\alpha_0^* = 0,$$
$$I^* = 0,$$
$$p^* = \frac{1}{2}(\frac{a}{b} + c),$$
$$\alpha_1^* = \alpha + \frac{b\left(\frac{a}{b}-c\right)^2}{8(K_{FT}+K_G)},$$

A proof of Theorem 4.7 is provided in Appendix A.2. Notice that the developer $F$'s decision on the foundation technology performance $\alpha_0^*$ always results in zero, indicating that the developer exits the game. Consequently, the deployer $S$ adopts a self-hosting approach. Interestingly, under a self-hosting solution, the unit price of the end technology, $p$, remains constant. This is due to the marginal cost of operations being fixed at $c$ and the end users' price sensitivity remaining stable at $b$. Additionally, as the hardware cost $K_G$ decreases, $S$ is incentivized to enhance the technology to a higher performance level, leading to an increase in $\alpha_1$ as the incremental advancement ($\alpha_1 - \alpha$) grows. Furthermore, the utilities of both the deployer, $U_S$, and the end users, $U_U$, increase.

THEOREM 4.8 (EXISTENCE OF SELF-HOSTING OUTCOME). *With a proactive O, given cost parameters $\{K_{FT}, K_{PRE}, c\}$ and market condition parameters $\{a, b\}$, there exists a threshold $\alpha^H \in \mathbb{R}^+$ such that $\forall K_G$, the game results in a self-hosting solution if $\alpha \in (\alpha^H, +\infty)$.*

A proof of Theorem 4.8 is provided in Appendix A.6. The insight is that when $O$ adopts a proactive strategy and develops the open-source technology to a sufficiently high performance level, the developer $F$ may initially be able to incentivize the deployer by either enhancing the closed-source technology or lowering the API price. During this process, profit gradually transfers from the developer to the deployer. However, as the performance of the open-source technology continues to increase, a point is reached where the developer can no longer offer enough incentives to attract the deployer while still ensuring its own profitability. Consequently, if the open-source technology achieves a high enough performance level, the closed-source developer foresees an unprofitable market and opts not to enter, ultimately resulting in a self-hosting outcome.

Besides, we examine the impact of proactive open-source engagement on technology outcomes using numerical results. We set

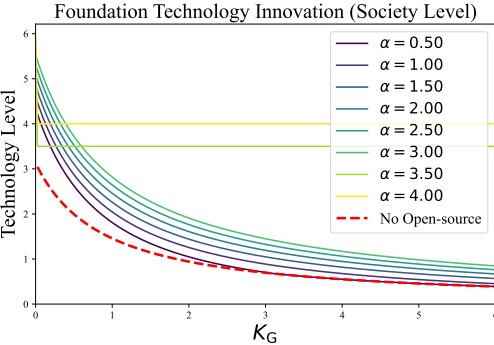

**(a) Foundation Technology**

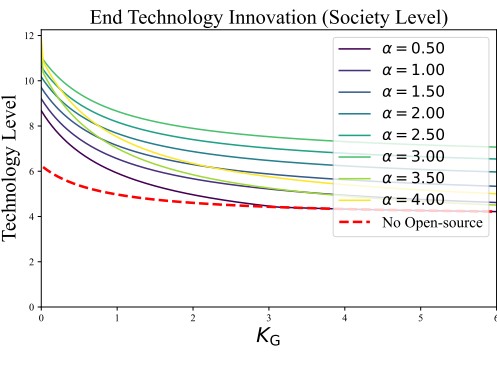

**(b) End Technology**

**Figure 3: Comparison of Technology Outcomes - No Open-source vs. Proactive Open-source Engagement ($a = 8$, $b = 1$, $c = 0.5$, $K_{FT} = K_{PRE} = 1$)**

parameters as follows: $a = 8$, $b = 1$, $c = 0.5$, $K_{FT} = K_{PRE} = 1$, and let $\alpha$ range from 0.5 to 4.

As shown in Figure 3, proactive open-source engagement leads to an increase in both foundation and end technology performance levels. The intuition is that when the open-source community independently sets the open-source technology level, rather than adjusting to closed-source performance, the closed-source developer cannot deter open-source technology by strategically reducing performance. Instead, the developer enhances the closed-source technology and lowers the API price to attract deployers to use the closed-source API, ultimately benefiting end technology innovation as well.

However, two horizontal lines appear in Figure 3a at $\alpha = 3.5$ and $\alpha = 4$, respectively. This indicates that, at these levels, open-source technology becomes advanced enough to drive the closed-source developer out of the market.

## 5 Impact of Open-source Engagement on Foundation Technology Innovation

In the above section, we show the impact of open-source engagement with numerical results under specified demand and cost functions. In this section, we examines more general situation. First, we

define the class of utility functions to which our conclusions apply. Then, we formally state the conditions under which open-source engagement may either encourage or hinder foundation technology innovation.

## 5.1 Concave and Unimodal Utility

First, we introduce two assumptions on the utility functions.

*Definition 5.1 (Strictly Unimodal Function).* A function $f : \mathbb{R} \times \mathbb{R} \to \mathbb{R}$ is called strictly unimodal over $x$ and $y$ if there exists a value $m \in D \subset \mathbb{R}$ such that $f$ is strictly increasing for $x \leq m$ and strictly decreasing for $x \geq m$, and there exists a value $n \in D \subset \mathbb{R}$ such that $f$ is strictly increasing for $y \leq n$ and strictly decreasing for $y \geq n$.

**Assumption 1**: The developer's utility $U_F(\alpha_1^*, p^*)$ is strictly concave in $\alpha_0$ and $\gamma$; that is, $\frac{\partial^2 U_F}{\partial \alpha_0^2} < 0$ and $\frac{\partial^2 U_F}{\partial \gamma^2} < 0$.

**Assumption 2**: The developer's utility $U_F(\alpha_1^*, p^*)$ is strictly unimodal in $\alpha_0$ and $\gamma$. This implies there exists a maximum utility at some values of $\alpha_0$ and $\gamma$ over their respective ranges.

Note: that the analysis in Section 4 satisfies these assumptions, ensuring that our conclusions hold within that framework.

## 5.2 Foundation Technology Innovation

Here, we formally state the theorems identifying the conditions under which open-source community engagement enhances or hinders foundation technology innovation.

THEOREM 5.2. *Assume the developer's strategy under no open-source engagement be characterized by $\alpha_0^*$ and $\gamma^*$, resulting in utility $U_S^{api}(\alpha_0^*, \gamma^*)$. After the engagement of a reactive open-source community, suppose the developer's strategy shifts to $\alpha_0'^*$ and $\gamma'^*$, yielding utility $U_S^{self}(\alpha_0'^*, \gamma'^*)$.*

*The developer's equilibrium technology level decreases, $\alpha_0'^* < \alpha_0^*$, if the following conditions hold:*

$$\frac{\partial U_S^{api}}{\partial \alpha_0} < \frac{\partial U_S^{self}}{\partial \alpha_0} \, and \, \frac{\partial^2 U_F}{\partial \gamma \partial \alpha_0} > 0.$$

*Conversely, the developer's equilibrium technology level increases, $\alpha_0'^* > \alpha_0^*$, if:*

$$\frac{\partial U_S^{api}}{\partial \alpha_0} > \frac{\partial U_S^{self}}{\partial \alpha_0} \quad and \quad \frac{\partial^2 U_F}{\partial \gamma \, \partial \alpha_0} < 0.$$

THEOREM 5.3. *Assume the developer's strategy under no open-source engagement be characterized by $\alpha_0^*$ and $\gamma^*$, resulting in utility $U_S^{self}(\alpha_0^*, \gamma^*)$. After proactive engagement by the open-source community, suppose the developer's strategy shifts to $\alpha_0'^*$ and $\gamma'^*$.*

*Then $\alpha_0'^* > \alpha_0^*$ if the following condition holds:*

$$\frac{\partial^2 U_F}{\partial \gamma \partial \alpha_0} < 0.$$

The proofs for Theorem 5.2 and Theorem 5.3 are provided in Appendix A.7 and Appendix A.8 respectively. Note that these conditions are sufficient but not necessary for the outcomes stated.

These theorems highlight the difference and similarity of reactive and proactive open-source engagement influencing foundation technology innovation:

- **Reactive Open-source Engagement**: When the deployer's utility gain from using the API is less sensitive to $\alpha_0$ than the self-hosting utility, the developer finds it challenging to achieve an advantage over open-source competition by enhancing technology performance. In such cases, the developer may strategically reduce the open-source competitiveness by lowering the technology performance, which in turn decreases the developer's utility. Due to the positive interaction between $\alpha_0$ and $\gamma$ in closed-source models, lowering $\gamma$ can help mitigate the rate of utility decline resulting from reduced technology performance. Additionally, a lower API price incentivizes deployers by reducing the cost associated with the API choice. Conversely, if the developer observes that the deployer's utility from the API is highly sensitive to $\alpha_0$ compared to self-hosting utility, a 'race-to-the-top' scenario arises where the developer is motivated to innovate more aggressively. In this case, while technology performance increases, the developer's utility may still decline. With a negative interaction between $\alpha_0$ and $\gamma$, lowering $\gamma$ helps counteract the rate of utility loss from potential over-innovation. A reduced API price further encourages deployers to adopt the API option.
- **Proactive Open-source Engagement**: In scenarios of proactive open-source engagement, the developer is consistently motivated to enhance technology performance to maintain an advantage over open-source alternatives, resulting in a continuous 'race-to-the-top.' Here, as technology performance improves, the developer's utility may experience diminishing returns. Given the negative interaction between $\alpha_0$ and $\gamma$, lowering $\gamma$ can help reduce the rate of utility decline associated with high levels of technology performance. Furthermore, a lower API price incentivizes deployers to continue using the API, aligning both parties' incentives towards higher technology standards.

## 6 Conclusion

This paper proposes a theoretical model that analyzes the interactions among closed-source developers, open-source communities, and deployers in the context of AI deployment. By examining three scenarios—no engagement, proactive open-source engagement, and reactive engagement—the model highlights how different open-source strategies can significantly shape deployment outcomes and drive innovation trajectories. These findings are particularly valuable for all stakeholders in the AI market, especially for the open-source community and regulatory bodies, as they provide insights into how various engagement strategies can either promote or inhibit technological progress.

Future research could build on this model by investigating additional factors such as the diverse motivations within open-source communities, the competitive dynamics in end markets, and the unique requirements across different deployer domains. We believe that societal outcomes are essential in shaping the technology market, and formalizing these considerations through theoretical models can provide a more comprehensive view of the AI ecosystem and help guide balanced and sustainable innovation strategies.

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

## A Proofs

### A.1 Proof of Theorem 4.1

As $I = 1$, the utility functions of are:

$$U_S^{api} = (p - \gamma c)(a - bp)\alpha_1 - K_S^{api}(\alpha_1 - \alpha_0)^2,$$

$$U_F^{api} = (\gamma c - c)(a - bp)\alpha_1 - K_F \alpha_0^2$$

where:

$$K_S^{api} = K_{FT}, \quad K_F^{api} = K_{PRE} + K_G.$$

**Step 1: Utility Maximization of $S$ for a Fixed $\alpha_0$ and $\gamma$**

$$\frac{\partial U_S^{api}}{\partial p} = (a + bc\gamma - 2bp)\alpha_1 = 0$$

$$\Rightarrow \quad p^* = \frac{\frac{a}{b} + c\gamma}{2}.$$

$$\frac{\partial U_S^{api}}{\partial \alpha_1} = (p - \gamma c)(a - bp) - 2K_S^{api}(\alpha_1 - \alpha_0) = 0.$$

Substituting $p^* = \frac{\frac{a}{b} + c\gamma}{2}$, we get:

$$\alpha_1^* = \alpha_0 + \frac{b\left(\frac{a}{b} - c\gamma\right)^2}{8K_{FT}}.$$

The optimal choices for $S$ are therefore:

$$p^* = \frac{\frac{a}{b} + c\gamma}{2}, \quad \alpha_1^* = \alpha_0 + \frac{b\left(\frac{a}{b} - c\gamma\right)^2}{8K_{FT}}.$$

**Step 2: Utility Maximization of $F$ Based on $S$'s Response**

Substituting $p^* = \frac{\frac{a}{b} + c\gamma}{2}$ and $\alpha_1^* = \alpha_0 + \frac{b\left(\frac{a}{b} - c\gamma\right)^2}{8K_{FT}}$, we have:

$$U_F^{api} = \frac{1}{2}bc^2(\gamma - 1)(\theta - \gamma)\left(\alpha_0 + \frac{bc^2(\theta - \gamma)^2}{8K_{FT}}\right) - K_F \alpha_0^2,$$

$$U_S^{api} = \frac{1}{4}bc^2(\theta - \gamma)^2 \alpha_0 + \frac{b^2 c^4}{64 K_{FT}}(\theta - \gamma)^4,$$

, where $\theta = \frac{a}{bc}$

$$\frac{\partial U_F^{api}}{\partial \alpha_0} = \frac{1}{2}(\gamma - 1)c(a - bc\gamma) - 2K_F \alpha_0 = 0$$

$$\Rightarrow \quad \alpha_0^* = \frac{(\gamma - 1)c(a - bc\gamma)}{4K_F}.$$

$$0 = \frac{\partial U_F^{api}}{\partial \gamma} = \frac{1}{2}bc^2\left(\alpha_0^*(\theta + 1 - 2\gamma^*) + \frac{bc^2}{8K_{FT}}(\theta - \gamma^*)^2(3 + \theta - 4\gamma^*)\right),$$

$$\Rightarrow \quad 0 = eq_a \cdot \gamma^2 + eq_b \cdot \gamma + eq_c,$$

where:

$$eq_a = 4(1 - \beta),$$

$$eq_b = 2\beta(3 + \theta) - 3 - 5\theta,$$

$$eq_c = \theta^2 + 3\theta - 2\beta(1 + \theta),$$

$$\beta = \frac{K_{FT}}{K_{PRE} + K_G}.$$

The discriminant $\delta$ is given by:

$$\delta = eq_b^2 - 4 \cdot eq_a \cdot eq_c.$$

Solving for the optimal $\gamma$ using the quadratic formula, we find:

$$\gamma^* = \frac{-eq_b - \sqrt{\delta}}{2 \cdot eq_a} = \frac{5\theta + 3 - 2\beta(3 + \theta) - \sqrt{\delta}}{8(1 - \beta)}.$$

### A.2 Proof of Theorem 4.7

When $I = 0$ and with a proactive open-source community, the utility functions for the deployer $S$ and the developer $F$ are given by:

$$U_S^{self} = (p - c)(a - bp)\alpha_1 - K_S^{self}(\alpha_1 - \alpha)^2,$$

$$U_F^{self} = 0,$$

where $K_S^{self} = K_{FT} + K_G$.

**Step 1: Solving for Optimal $p^*$ and $\alpha_1^*$ for $S$**

$$\frac{\partial U_S^{self}}{\partial p} = (a - 2bp + bc)\alpha_1 = 0,$$

$$\Rightarrow \quad p^* = \frac{1}{2}\left(\frac{a}{b} + c\right).$$

$$\frac{\partial U_S^{self}}{\partial \alpha_1} = (p - c)(a - bp) - 2K_S^{self}(\alpha_1 - \alpha) = 0.$$

Substituting $p^* = \frac{1}{2}\left(\frac{a}{b} + c\right)$,

$$\alpha_1^* = \alpha + \frac{b\left(\frac{a}{b} - c\right)^2}{8(K_{FT} + K_G)}.$$

Thus, the optimal utility for $S$ in a self-hosting setup with proactive $O$ is:

$$U_S^{self}(\alpha_1^*, p^*) = \frac{1}{4}bc^2(\theta - 1)^2 \alpha + \frac{b^2 c^4}{64(K_{FT} + K_G)}(\theta - 1)^4$$

, where $\theta = \frac{a}{bc}$.

**Step 2: Confirming Developer's Choice (Setting $I = 0$)**

Since $U_F^{self} = 0$ when $S$ chooses self-hosting, the developer $F$ gains no utility. This setup implies that the optimal strategy for the developer is to exit the market, yielding:

$$\gamma^* = 1, \quad \alpha_0^* = 0, \quad I^* = 0.$$

### A.3 Proof of Theorem 4.3

From A.2, we have:

$$U_S^{self}(\alpha_1^*, p^*) = \frac{1}{4}bc^2(\theta - 1)^2 \alpha + \frac{b^2 c^4}{64(K_{FT} + K_G)}(\theta - 1)^4.$$

In this API case, $\alpha = m\alpha_0$, thus:

$$U_S^{self}(\alpha_1^*, p^*) = \frac{1}{4}bc^2(\theta - 1)^2 m\alpha_0 + \frac{b^2 c^4}{64(K_{FT} + K_G)}(\theta - 1)^4.$$

From Section A.1, we have:

$$U_S^{api}(\alpha_1^*, p^*) = \frac{1}{4}bc^2(\theta - \gamma)^2 \alpha_0 + \frac{b^2 c^4}{64 K_{FT}}(\theta - \gamma)^4$$

The goal is to solve:

$$\alpha_0^*, \gamma^* = \arg\max_{\alpha_0, \gamma} U_F^{api}(\alpha_1^*, p^*),$$

subject to:

$$U_S^{self}(\alpha_1^*, p^*) \leq U_S^{api}(\alpha_1^*, p^*). \tag{26}$$

Define the Lagrangian with multiplier $\lambda$:

$$\mathcal{L} = U_F^{api}(\alpha_1^*, p^*) + \lambda \left( U_S^{api}(\alpha_1^*, p^*) - U_S^{self}(\alpha_1^*, p^*) \right).$$

$$\begin{cases} \frac{\partial \mathcal{L}}{\partial \alpha_0} = 0, \\ \frac{\partial \mathcal{L}}{\partial \gamma} = 0, \\ \frac{\partial \mathcal{L}}{\partial \lambda} = 0 \end{cases}$$

Thus,

$$\begin{cases} \frac{\partial U_F^{api}}{\partial \alpha_0} + \lambda(\frac{\partial U_S^{api}}{\partial \alpha_0} - \frac{\partial U_S^{self}}{\partial \alpha_0}) = 0, \\ \frac{\partial U_F^{api}}{\partial \gamma} + \lambda(\frac{\partial U_S^{api}}{\partial \gamma} - \frac{\partial U_S^{self}}{\partial \gamma}) = 0, \\ U_S^{self}(\alpha_1^*, p^*) = U_S^{api}(\alpha_1^*, p^*) \end{cases}$$

Thus,

$$\begin{cases} \frac{\partial U_F^{api}}{\partial \alpha_0}(\frac{\partial U_S^{api}}{\partial \gamma} - \frac{\partial U_S^{self}}{\partial \gamma}) = \frac{\partial U_F^{api}}{\partial \gamma}(\frac{\partial U_S^{api}}{\partial \alpha_0} - \frac{\partial U_S^{self}}{\partial \alpha_0}), \\ U_S^{self}(\alpha_1^*, p^*) = U_S^{api}(\alpha_1^*, p^*) \end{cases}$$

The partial derivatives of $U_S^{api}$ and $U_S^{self}$ are as follows:

$$\begin{cases} \frac{\partial U_S^{api}}{\partial \alpha_0} = \frac{1}{4}bc^2(\theta - \gamma)^2, \\ \frac{\partial U_S^{api}}{\partial \gamma} = -\frac{b^2 c^4}{16 K_{FT}}(\theta - \gamma)^3 - \frac{1}{2}bc^2(\theta - \gamma)\alpha_0, \\ \frac{\partial U_S^{self}}{\partial \alpha_0} = \frac{1}{4}mbc^2(\theta - 1)^2, \\ \frac{\partial U_S^{self}}{\partial \gamma} = 0, \\ \frac{\partial U_F^{api}}{\partial \alpha_0} = \frac{1}{2}(\gamma - 1)c(a - bc\gamma) - 2K_F \alpha_0, \\ \frac{\partial U_F^{api}}{\partial \gamma} = \frac{1}{2}bc^2 \left( \alpha_0^*(\theta + 1 - 2\gamma^*) + \frac{bc^2}{8 K_{FT}}(\theta - \gamma^*)^2(3 + \theta - 4\gamma^*) \right) \end{cases}$$

The optimal values $\alpha_0^*$ and $\gamma^*$ satisfy:

$$\begin{cases} \left(16(\theta - \gamma^*)^2 - m(\theta - 1)^2\right)\alpha_0^* = \left(\frac{(\theta-1)^4}{K_{FT}+K_G} - \frac{(\theta-\gamma^*)^4}{K_{FT}}\right)bc^2, \\ \frac{2\left(bc^2(\gamma^*-1)(\theta-\gamma^*) - 4(K_{PRE}+K_G)\alpha_0^*\right)}{\alpha_0^*(\theta+1-2\gamma^*) + \frac{bc^2}{8K_{FT}}(\theta-\gamma^*)^2(3+\theta-4\gamma^*)} = \frac{-bc^2\left((\theta-\gamma^*)^2 - m(\theta-1)^2\right)}{\frac{bc^2}{8K_{FT}}(\theta-\gamma^*)^3 + (\theta-\gamma)\alpha_0^*}, \end{cases}$$

where $\theta = \frac{a}{bc}$.

## A.4 Proof of Theorem 4.6

From A.2, with proactive open-source community, we have:

$$U_S^{self}(\alpha_1^*, p^*) = \frac{1}{4}bc^2(\theta - 1)^2 \alpha + \frac{b^2 c^4}{64(K_{FT}+K_G)}(\theta - 1)^4.$$

Same as A.3, the goal is to solve:

$$\alpha_0^*, \gamma^* = \arg\max_{\alpha_0, \gamma} U_F^{api}(\alpha_1^*, p^*),$$

subject to:

$$U_S^{self}(\alpha_1^*, p^*) \le U_S^{api}(\alpha_1^*, p^*).$$

Similar as A.3, we can solve the problem with KKT. The only difference is that $\frac{\partial U_S^{self}}{\partial \alpha_0} = 0$

## A.5 Proof of Theorem 4.4

To prove Theorem 4.4, we need to find $\alpha_0^*, \gamma^*$ satisfying:

$$\begin{cases} U_S^{self}(\alpha_1^*, p^*) \le U_S^{api}(\alpha_1^*, p^*), \\ U_F^{api} \ge 0, \end{cases}$$

From A.1 and A.3, we know it is equivalent to find $\alpha_0^*, \gamma^*$ satisfying:

$$\begin{cases} \left(16(\theta - \gamma^*)^2 - m(\theta - 1)^2\right)\alpha_0^* > \left(\frac{(\theta-1)^4}{K_{FT}+K_G} - \frac{(\theta-\gamma^*)^4}{K_{FT}}\right)bc^2, \\ \frac{1}{2}bc^2(\gamma - 1)(\theta - \gamma)\left(\alpha_0 + \frac{bc^2(\theta-\gamma)^2}{8K_{FT}}\right) - K_F \alpha_0^2 > 0 \end{cases}$$

, where $\theta = \frac{a}{bc}$.

Also, reasonable solution should satisfy $\alpha_0^* > 0$ and $1 < \gamma^* < \theta$.

$U_F^{api}(\alpha_0)$ is a quadratic function, opening downward, with $U_F^{api}(\alpha_0 = 0) > 0$ and axis of symmetry given by:

$$\frac{bc^2}{4K_F}(\gamma - 1)(\theta - \gamma) > 0,$$

Define $\alpha_0^1 = \frac{bc^2}{4K_F}(\gamma - 1)(\theta - \gamma)$.

### A.5.1 Case 1: $m \ge 1$.

$$(\theta - \gamma)^2 - m(\theta - 1)^2 < 0.$$

Thus, we must have:

$$\frac{(\theta - 1)^4}{K_{FT} + K_G} < \frac{(\theta - \gamma^*)^4}{K_{FT}} \Rightarrow (\theta - \gamma)^4 > \left(\frac{K_{FT}}{K_{FT} + K_G}\right)^{1/4} \cdot (\theta - 1).$$

We can always find $\gamma^*$ to satisfy this condition.

Also,

$$\alpha_0 < \frac{\left(\frac{(\theta-1)^4}{K_{FT}+K_G} - \frac{(\theta-\gamma^*)^4}{K_{FT}}\right)bc^2}{16(\theta - \gamma^*)^2 - m(\theta - 1)^2} = \alpha_0^{cut}.$$

Thus, an example solution:

$$\begin{cases} \gamma^* = \theta - \frac{1}{2}\left(\theta - 1 + \left(\frac{K_{FT}}{K_{FT}+K_G}\right)^{1/4} \cdot (\theta - 1)\right), \\ \alpha_0^* = \min\{\alpha_0^1, \alpha_0^{cut}\}. \end{cases}$$

### A.5.2 Case 2: $0 < m < 1$. .

If $0 \le m^2 < \frac{K_{FT}}{K_{FT}+K_G} < 1$ (when hardware cost is relatively low), let

$$\gamma^* \in \left(1, \theta - \left(\frac{K_{FT}}{K_{FT}+K_G}\right)^{1/4} \cdot (\theta - 1)\right),$$

which ensures:

$$\begin{cases} (\theta - \gamma)^2 - m(\theta - 1)^2 > 0, \\ \frac{(\theta-1)^4}{K_{FT}+K_G} - \frac{(\theta-\gamma^*)^4}{K_{FT}} < 0. \end{cases}$$

An example solution:

$$\begin{cases} \gamma^* = \theta - \frac{1}{2}\left(\theta - 1 + \left(\frac{K_{FT}}{K_{FT}+K_G}\right)^{1/4} \cdot (\theta - 1)\right), \\ \alpha_0^* = \alpha_0^1. \end{cases}$$

If $\frac{K_{FT}}{K_{FT}+K_G} \leq m^2 < 1$ (when hardware cost is relatively high), let

$$\gamma^* \in \left(1, \theta - m^{1/2} \cdot (\theta - 1)\right),$$

which ensures:

$$\begin{cases} (\theta - \gamma)^2 - m(\theta - 1)^2 > 0, \\ \frac{(\theta-1)^4}{K_{FT}+K_G} - \frac{(\theta-\gamma^*)^4}{K_{FT}} < 0. \end{cases}$$

An example solution:

$$\begin{cases} \gamma^* = \theta - \frac{1}{2}\left(\theta - 1 + m^{1/2} \cdot (\theta - 1)\right), \\ \alpha_0^* = \alpha_0^1. \end{cases}$$

## A.6 Proof of Theorem 4.8

Contrary A.5, we need to illustrate: when $\alpha$ is high, there is no solution of $\alpha_0^*, \gamma^*$ satisfying:

$$\begin{cases} 16(\theta - \gamma^*)^2 \alpha_0^* - 16(\theta - 1)^2\alpha > \left(\frac{(\theta-1)^4}{K_{FT}+K_G} - \frac{(\theta-\gamma^*)^4}{K_{FT}}\right)bc^2, \\ U_F^{api} = \frac{1}{2}bc^2(\gamma - 1)(\theta - \gamma)\left(\alpha_0 + \frac{bc^2(\theta-\gamma)^2}{8K_{FT}}\right) - K_F\alpha_0^2 > 0 \end{cases}$$

, where $\theta = \frac{a}{bc}$.

It is equivalent to:

$$\begin{cases} 16(\theta - \gamma^*)^2 \alpha_0^* > \left(\frac{(\theta-1)^4}{K_{FT}+K_G} - \frac{(\theta-\gamma^*)^4}{K_{FT}}\right)bc^2 + 16(\theta - 1)^2\alpha, \\ \frac{1}{2}bc^2(\gamma - 1)(\theta - \gamma)\left(\alpha_0 + \frac{bc^2(\theta-\gamma)^2}{8K_{FT}}\right) - K_F\alpha_0^2 > 0 \end{cases}$$

Denote $R = \left(\frac{(\theta-1)^4}{K_{FT}+K_G} - \frac{(\theta-\gamma^*)^4}{K_{FT}}\right)bc^2 + 16(\theta-1)^2\alpha$, $L = 16(\theta-\gamma^*)^2$
Thus,

$$\begin{cases} L\alpha_0^* > R, \\ \frac{1}{2}bc^2(\gamma - 1)(\theta - \gamma)\left(\alpha_0 + \frac{bc^2(\theta-\gamma)^2}{8K_{FT}}\right) - K_F\alpha_0^2 > 0 \end{cases}$$

Lets $\alpha > \frac{(\theta-1)^4}{K_{FT}}bc^2$, thus,

$$\begin{cases} \alpha_0^* > \frac{R}{L} > 0., \\ \frac{1}{2}bc^2(\gamma - 1)(\theta - \gamma)\left(\alpha_0 + \frac{bc^2(\theta-\gamma)^2}{8K_{FT}}\right) - K_F\alpha_0^2 > 0 \end{cases}$$

As $U_F^{api}(\alpha_0)$ is a quadratic function, opening downward, with $U_F^{api}(\alpha_0 = 0) > 0$ and axis of symmetry given by:

$$\frac{bc^2}{4K_F}(\gamma - 1)(\theta - \gamma) > 0,$$

We only need to substitute $\alpha_0^* = \frac{R}{L}$ in $U_F^{api}(\alpha_0*)$ and show $U_F^{api}(\alpha_0) < 0$ when $\alpha$ is high.

$$U_F^{api}\left(\alpha_0^* = \frac{R}{L}\right) = -K_F\left(\frac{R}{L}\right)^2 + \frac{bc^2}{2}(\gamma-1)(\theta-1)\frac{R}{L} + \frac{b^2c^4}{16K_{FT}}(\theta-\gamma)^3(\gamma-1)$$

$$= (-K_FR^2 + \frac{bc^2}{2}(\gamma-1)(\theta-1)RL + \frac{b^2c^4}{16K_{FT}}(\theta-\gamma)^3(\gamma-1)L^2)/L^2.$$

$$\begin{cases} R \geq R_{\min} = R(\gamma^* = 1) = \left(\frac{(\theta-1)^4}{K_{FT}+K_G} - \frac{(\theta-1)^4}{K_{FT}}\right)bc^2 + 16(\theta-1)^2\alpha, \\ R \leq R - bc^2\frac{(\theta-1)^4}{K_{FT}+K_G} = -\frac{(\theta-\gamma^*)^4}{K_{FT}}bc^2 + 16(\theta-1)^2\alpha, \\ L \leq L_{\max} = L(\gamma^* = 1) = 16(\theta-1)^2, \\ \theta - \gamma \leq \theta - 1. \end{cases}$$

Thus,

$$L^2U_F^{api}\left(\alpha_0^* = \frac{R}{L}\right) \leq -K_F\left(\left(\frac{(\theta-1)^4}{K_{FT}+K_G} - \frac{(\theta-\gamma^*)^4}{K_{FT}}\right)bc^2 + 16(\theta-1)^2\alpha\right)^2$$

$$+ \frac{bc^2}{2}(\gamma-1)(\theta-1)\left(-\frac{(\theta-\gamma^*)^4}{K_{FT}}bc^2 + 16(\theta-1)^2\alpha\right)16(\theta-1)^2$$

$$+ \frac{b^2c^4}{16K_{FT}}(\theta-1)^3(\gamma-1)16^2(\theta-1)^4.$$

Obviously, $L^2U_F^{api}\left(\alpha_0^* = \frac{R}{L}\right)$ is a quadratic function of $\alpha$, opening downward. Thus, when $\alpha$ is high enough to let $L^2U_F^{api}\left(\alpha_0^* = \frac{R}{L}\right) < 0$, we cannot find a solution of $\alpha_0^*, \gamma^*$ to make $U_F^{api} > 0$, which the equilibrium falls into self-hosting. The cut-off $\alpha^H$ can be the right root of $L^2U_F^{api}\left(\alpha_0^* = \frac{R}{L}\right) = 0$

## A.7 Proof of Theorem 5.2

From A.2, we know $\alpha_0'^*$ and $\gamma'^*$ is the solution of

$$\begin{cases} \frac{\partial U_F^{api}}{\partial\alpha_0} + \lambda\left(\frac{\partial U_S^{api}}{\partial\alpha_0} - \frac{\partial U_S^{self}}{\partial\alpha_0}\right) = 0, \\ \frac{\partial U_F^{api}}{\partial\gamma} + \lambda\frac{\partial U_S^{api}}{\partial\gamma} = 0, \\ U_S^{self} = U_S^{api} \end{cases}$$

However, $\alpha_0^*$ and $\gamma^*$ is the solution of:

$$\begin{cases} \frac{\partial U_F^{api}}{\partial\alpha_0} = 0, \\ \frac{\partial U_F^{api}}{\partial\gamma} = 0, \end{cases}$$

Also, we know that $U_S^{self}(\alpha_0^*, \gamma^*) > U_S^{api}(\alpha_0^*, \gamma^*)$. Else it falls into a API-dominant solution.

**Part One:** if

$$\frac{\partial U_S^{api}}{\partial\alpha_0} < \frac{\partial U_S^{self}}{\partial\alpha_0} \, and \, \frac{\partial^2 U_F}{\partial\gamma\partial\alpha_0} > 0. \tag{27}$$

We know

$$\begin{cases} \frac{\partial U_F^{api}}{\partial\alpha_0}(\gamma'^*, \alpha_0'^*) > 0, \\ \frac{\partial U_F^{api}}{\partial\gamma}(\gamma'^*, \alpha_0'^*) > 0 \end{cases}$$

We discuss the value of $\alpha_0'^*$ and $\gamma'^*$:

(1) $\alpha_0'^* > \alpha_0^*, \gamma'^* > \gamma^*$: unreasonable. As $U_S^{self}(\alpha_0'^*, \gamma'^*) - U_S^{api}(\alpha_0'^*, \gamma'^*)$
$> U_S^{self}(\alpha_0'^*, \gamma^*) - U_S^{api}(\alpha_0'^*, \gamma^*) > U_S^{self}(\alpha_0^*, \gamma^*) - U_S^{api}(\alpha_0^*, \gamma^*) > 0$

(2) $\alpha_0'^* > \alpha_0^*$, $\gamma'^* < \gamma^*$: unreasonable. As $\frac{\partial U_F^{api}}{\partial \alpha_0}(\gamma'^*, \alpha_0'^*) < \frac{\partial U_F^{api}}{\partial \alpha_0}(\gamma^*, \alpha_0'^*) < \frac{\partial U_F^{api}}{\partial \alpha_0}(\gamma^*, \alpha_0^*) = 0$

(3) $\alpha_0'^* < \alpha_0^*$, $\gamma'^* > \gamma^*$: unreasonable. As $\frac{\partial U_F^{api}}{\partial \gamma}(\gamma'^*, \alpha_0'^*) < \frac{\partial U_F^{api}}{\partial \gamma}(\gamma^*, \alpha_0'^*) < \frac{\partial U_F^{api}}{\partial \gamma}(\gamma^*, \alpha_0^*) = 0$

Thus, $\alpha_0'^* < \alpha_0^*$, $\gamma'^* < \gamma^*$ is the only feasible solution.

**Part Two:** if

$$\frac{\partial U_S^{api}}{\partial \alpha_0} > \frac{\partial U_S^{self}}{\partial \alpha_0} \quad and \quad \frac{\partial^2 U_F}{\partial \gamma \partial \alpha_0} < 0. \tag{28}$$

We know

$$\begin{cases} \frac{\partial U_F^{api}}{\partial \alpha_0}(\gamma'^*, \alpha_0'^*) < 0, \\ \frac{\partial U_F^{api}}{\partial \gamma}(\gamma'^*, \alpha_0'^*) > 0 \end{cases}$$

We discuss the value of $\alpha_0'^*$ and $\gamma'^*$:

(1) $\alpha_0'^* > \alpha_0^*$, $\gamma'^* > \gamma^*$: unreasonable. As $\frac{\partial U_F^{api}}{\partial \gamma}(\gamma'^*, \alpha_0'^*) < \frac{\partial U_F^{api}}{\partial \gamma}(\gamma^*, \alpha_0'^*) < \frac{\partial U_F^{api}}{\partial \gamma}(\gamma^*, \alpha_0^*) = 0$

(2) $\alpha_0'^* < \alpha_0^*$, $\gamma'^* < \gamma^*$: unreasonable. As $\frac{\partial U_F^{api}}{\partial \alpha_0}(\gamma'^*, \alpha_0'^*) > \frac{\partial U_F^{api}}{\partial \alpha_0}(\gamma^*, \alpha_0'^*) > \frac{\partial U_F^{api}}{\partial \alpha_0}(\gamma^*, \alpha_0^*) = 0$

(3) $\alpha_0'^* < \alpha_0^*$, $\gamma'^* > \gamma^*$: unreasonable. As $U_S^{self}(\alpha_0'^*, \gamma'^*) - U_S^{api}(\alpha_0'^*, \gamma'^*) > U_S^{self}(\alpha_0'^*, \gamma^*) - U_S^{api}(\alpha_0'^*, \gamma^*) > U_S^{self}(\alpha_0^*, \gamma^*) - U_S^{api}(\alpha_0^*, \gamma^*) > 0$

Thus, $\alpha_0'^* > \alpha_0^*$, $\gamma'^* < \gamma^*$ is the only feasible solution.

## A.8 Proof of Theorem 5.3

The proof is the same as the Part Two of A.7

# B Figures

## B.1 Equilibrium Without Open-source Engagement

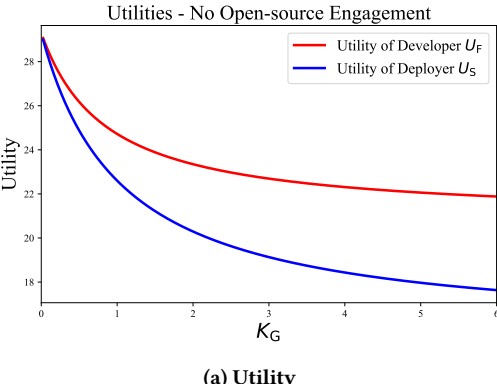

(a) Utility

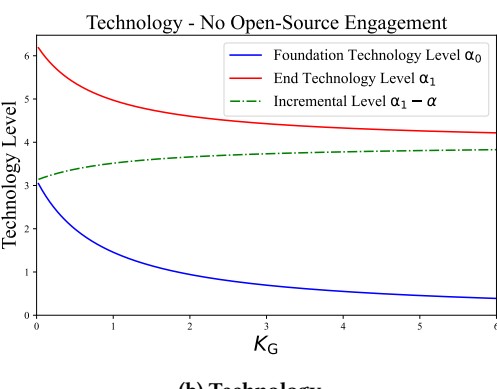

(b) Technology

Figure 4: Equilibrium Outcomes without Open-source Engagement ($a = 8, b = 1, c = 0.5, K_{FT} = K_{PRE} = 1$)

# C Notations

**Table 1: Notations**

| Symbols | Meanings |
| --- | --- |
| $F$ | closed-source foundation technology developer |
| $S$ | Domain-specific deployer |
| $O$ | Open-source community |
| $U$ | End user |
| $a$ | Total potential demand in the market |
| $b$ | Price sensitivity of end user |
| $\gamma$ | Price multiplier of API |
| $\alpha_0$ | Closed-source foundation technology performance |
| $\alpha_0^{soc}$ | Foundation technology performance at social level |
| $\alpha$ | Open-source foundational technology performance |
| $\alpha_1$ | End technology performance |
| $\alpha_1^{soc}$ | End technology performance at social level |
| $m$ | Relative performance of open-source to closed-source foundational technology |
| $c$ | Unit operation cost of the technology |
| $K_F$ | Cost factor for developing foundational technology |
| $K_S^{api}$ | Cost factor for adapting technology in API scenario |
| $K_S^{self}$ | Cost factor for adapting technology in self-hosting scenario |
| $K_{GPU}$ | Hardware cost parameter |
| $K_{PRE}$ | Non-hardware cost parameter for developing foundation technology |
| $K_{FT}$ | Non-hardware cost factor for adapting technology |
| $p$ | Price of domain-specific technology |
| $D$ | Actual demand in the end market |
| $U_S$ | Utility of deployer $S$ |
| $U_F$ | Utility of developer $F$ |

