# OpenReview forum: "Navigating the Deployment Dilemma and Innovation Paradox: Open-Source v.s. Closed-source Models"
_ACM.org/TheWebConf/2025/Conference — WWW 2025 Oral_

### Official Review · Reviewer_EG9x · 2024-11-29

**Novelty:** 4
**Technical Quality:** 3

**Review:**

Pros:

- Innovative topic and scope
- Modelling and approach to the study

Cons:

- Concerns about the plausibility of the hypotheses
- Complexity in the interpretation of the results
- Limited (and apparently very theoretical) numerical analysis
- Discussion of implications for policymakers
- Writing and presentation details

The paper presents a game-theoretical approach to the interplay between foundation model innovation and development (closed / open source), and deployment of those models to end customers. I find the topic interesting for the community, and I generally like the approach of the author(s) on analysing the topic. Still, I find the paper and the analysis needs some work to be acceptable in top venues for a number of reasons I will proceed to explain.

First, I have some concerns about the hypotheses of the study that I am listing as questions for the author(s) that impact heavily the results of the study. For example:

- The reactive open source community is able to achieve a performance $\alpha = m \cdot \alpha_0$, being m a constant. As a result, for m>1, closed-source developers F have incentives to reduce $\alpha$ to minimize the premium of the open source community O and reduce the cost of the API. However, I would say that this is not realistic, and that a low performance of F would leave more room for improvement to O, and hence lead to higher m. And viceversa, if F achieve a very high $\alpha$, it is more difficult for O to achieve m>1. What is the rationale behind $\alpha = m \cdot \alpha_0$? Is linear realistic here?
- Domain-specific deployers (S) can decide whether to host the open source technology or use the API, and regardless of this decision they are able to adapt the technology and achieve the same $\alpha_1$. Is not $\alpha_1$ dependent on their decision? I mean, is it realistic that S can achieve the same $\alpha_1$ with the API and with the open-access foundation model regardless their level of development?
- Also, in answering the previous question, take into account that another hypothesis is that unit costs of developing the API and increasing its performance is that it only involves software ($K_{FT}$), and the cost of increasing the performance of open source models will always be higher because it also involves hardware ($K_G$). This gives a cost advantage to increasing the performance of the API, which actually explains the strategic behavior of F. Is it realistic that assumption combined with the possibility that both decisions lead to the same performance? At the end, the use of API has also hardware cost, although only a tariff to API users. Assuming that closed-form developers can arbitrarily lower their price is assuming they have a cost advantage over O and S. My point is that at some point they will drop the price below cost (they also have hardware to run the API) and commit predatory pricing to make O exit the market.
- The paper assumes there is only competition between open-source and close-form developers, but there are several closed-form developers and several open-source models, each of them making their own decisions.

Second, the results of closed form are complex and difficult to interpret. I think the author(s) can try to simplify expressions and make a better job at interpreting them for the reader. The conclusion does not state clearly the conclusion: that proactive open-source engagement is desirable to create a competition in the market.

Third, the paper claims that one of the contributions is discussing "broader implications for policymakers, offering insights into how the regulation [...] can support sustainable AI innovation. I think that discussion in the paper is, at most, very limited and, in line with previous comments about the presentation of results, convoluted. For example, I miss connection with policy topics like whether and when policy makers shouold support open-source innovation in foundation models?

Generally, formatting can be improved. Figure 1 is too big and can be reduced, and Table 2 with the summary of the notation at the end of the appendix could be in the body of the paper. Figures ideally should be readable in B&W, consider for example adding bullets.

I would say the paper needs a thorough writing review and proof reading to be more understandable and generate more acute and meaningful findings. Some examples:

- In the abstract, I find the following sentence difficult to understand until you read the whole paper: "We find that the deployer consistently opts for closed source APIs when the open-source community engages in the market reactively by maintaining a fixed performance ratio relative to closed-source advancements."
- Check for missing citations (see introduction) and cross-references (Theorems)
- Incomplete sentences, see for instance this one in the introduction: "As the landscape of foundational models is characterized by two prominent alternatives: open-source and closed-source." That whole paragraph need full rewriting. Or check the sentence before Eq (18)
- Leave spaces before citations
- Check definition 3.1, is it correct or should it be "naturally satisfies $U_S^{self} \geq U_S^{api}?"
- Check for typos and weird expressions: "strategis", "it expect", "we specify the forms of solution", $U_{F,api}$ in definition 3.2
- Introduce the concept technology level before Figure 2. I assume it refers to $\alpha$, but this must be specified in the text.
- Orphan colon at the end of page 8

Finally, regarding references, avoid using ArXiv as possible and point to the equivalent journal or conference papers. For example, Ref 10 is Harvard Data Science Review, and ref 13 is NIPS.

**Questions:**

What is the rationale behind $\alpha = m \cdot \alpha_0$? Is linear realistic here?

Is not $\alpha_1$ dependent on their decision? I mean, is it realistic that S can achieve the same $\alpha_1$ with the API and with the open-access foundation model regardless their level of development?

Is it realistic the cost advantage of using the API combined with the possibility that this can lead to the same performance than using an open source model?

How do results apply to situations where there are several closed-form developers F, several open source models, and many deployers making their own decisions?

How realistic are the hypotheses in the numerical results ($𝑎 = 8, 𝑏 = 1, 𝑐 = 0.5,𝐾_{𝐹𝑇} = 𝐾_{𝑃𝑅𝐸} = 1$)?

**Reviewer Confidence:**

3: The reviewer is confident but not certain that the evaluation is correct

**Scope:**

3: The work is somewhat relevant to the Web and to the track, and is of narrow interest to a sub-community

---

### Official Review · Reviewer_mqSP · 2024-12-01

**Novelty:** 4
**Technical Quality:** 2

**Review:**

Pros:
* I like the problem of analyzing the game between model developers. This is particularly an interesting problem as more and more LLMs are developed from various sources.
* In spite of several limitations of the model (see Cons), I feel the most general version of the model makes sense as a starting point. That's saying, the model captures some interesting interactions between developers with diverse interests.

Cons:
* The main concern is about the model. The model is a little restrictive in many senses, which questions the usefulness of the key insights. I understand that several assumptions are necessary to get closed-form solutions, but the main concern is the generalizability of the insights. Here are several examples that bother me a little:
    * Who is the domain-specific developer in practice? I assume OpenAI is F. Then, users always have the option of using ChatGPT directly. This gives users an outside option which seems not modeled but quite important. In particular, the demand should not be monopolized by $p$ and $\alpha_1$, but is influenced by $\gamma$ and $\alpha_0$. Of course, this makes the analysis more complex. However, could the authors discuss whether this makes sense and whether the key insights can be generalized to these more practical settings?
    * The cost of operation of the open-source model and the API is the same.
    * The most interesting result (in my opinion) is Theorem 4.4, which suggests that if the open-source community's model performance is a scaler of the closed-source model, the closed-source model can always win the market. However, I feel this counterintuitive result is specifically caused by the simplicity of the model. The intuition of this result is that when m is large, the closed-source developer can strategically reduce the performance of its own model to harm the open-source model. I feel this is a little unrealistic. The result can be more motivating if the authors can connect this with practice, e.g. by showing some empirical evidence of the model developers' strategic behaviors.
    * I understand that the linear demand and quadratic cost models are used to develop close-form solutions, and they are common assumptions. However, these results lack certain generalizability. I would suggest using different models at least in simulations to test whether the key insights can be generalized, e.g. Theorem 4.4.

Minor points:
    * Is there a typo in definitions 3.1, 3.2, and 3.3? Shoudn't it be $U_S^{self} \le U_S^{api}$?
    * Many places that citation or reference cannot be found and showing ??
    * I'm curious about the citation format. For [25], Kleinberg is neither the first author nor the last author, while whenever cited, it shows "Kleinberg et al. [25]". Is this caused by a misuse of the package?

**Questions:**

See Review please.

**Reviewer Confidence:**

3: The reviewer is confident but not certain that the evaluation is correct

**Scope:**

3: The work is somewhat relevant to the Web and to the track, and is of narrow interest to a sub-community

---

### Official Review · Reviewer_SzLH · 2024-12-02

**Novelty:** 5
**Technical Quality:** 4

**Review:**

Summary

The authors examine the trade-offs organizations face when choosing between closed-source and open-source models. They propose a framework grounded on game theory to analyze three scenarios: no open-source engagement, proactive open-source engagement, and reactive open-source engagement.
They find that deployers prefer closed-source APIs in reactive open-source engagement and prefer open-source models in proactive open-source engagement. They also identify conditions under which engagement and competitiveness of the open-source community can foster or inhibit technological progress.


Pros
* The paper is easy to read and well-organized organized. It clearly defines the utility functions, assumptions, and scenarios.
* The paper addresses the economic aspect of deploying generative AI models, focusing on the cost implications of choosing between open-source and closed-source options.
* The game-theoretic model is mathematically rigorous and well-formulated.


Cons
* The conclusions rely on theoretical models without direct validation through empirical or experimental data. A validation plan or discussion can strengthen the paper.
* Since some readers may not be well-versed in AI and economics, a glossary of terms for readers unfamiliar with game theory or economic modeling would be helpful.


Others
* Page 1 Line 67, [25??]
* Page 2 Line 200, strategis --> strategies.
* Page 6 Line 608, Theorem ??

**Questions:**

* Have you considered other scenarios that are in between the three scenarios?

* How can the model be validated using real-world data?

* Have you considered the ethical implications of your findings, such as how increased reliance on closed-source APIs might impact data privacy, accessibility, and fairness?

**Reviewer Confidence:**

3: The reviewer is confident but not certain that the evaluation is correct

**Scope:**

3: The work is somewhat relevant to the Web and to the track, and is of narrow interest to a sub-community

---

### Official Review · Reviewer_hqi7 · 2024-12-02

**Novelty:** 4
**Technical Quality:** 4

**Review:**

This paper uses game theory to study how companies choose between open-source and closed-source AI models. It finds that companies prefer closed-source APIs when open-source communities just follow and copy, but may choose open-source when these communities actively innovate on their own.

Pros:
1. Well-structured theoretical framework with clear model specifications and assumptions
2. Rigorous mathematical analysis with detailed proofs
3. Clear definitions of key concepts

Cons:
1. Frequent typos, grammar issues and error in reference

2. Experimental validation relies solely on synthetic data

3. Not considering dynamic market changes

**Questions:**

1.How robust are the key assumptions in the model?

2.How can the model be extended to include more complex market dynamics?

3.How does the model account for temporal changes?

**Reviewer Confidence:**

2: The reviewer is willing to defend the evaluation, but it is likely that the reviewer did not understand parts of the paper

**Scope:**

3: The work is somewhat relevant to the Web and to the track, and is of narrow interest to a sub-community

---

### Official Review · Reviewer_htuR · 2024-12-03

**Novelty:** 5
**Technical Quality:** 5

**Review:**

The paper uses game theory to analyze the economic and innovation dynamics between open-source and closed-source AI models. It explores the trade-offs faced by deployers and the effects of proactive or reactive strategies by the open-source community on market results and technological progress.

However, the paper has several typos and missing references, which affect its clarity. For example:
1. Line 67: Missing reference (development [25 ? ?]).
2. Line 608: Missing reference (Theorem ??).

> Strengths of the Paper:

1. Applying game theory to study deployment choices and innovation dynamics is a fresh and insightful approach.
2. The paper presents a detailed multi-stage game model with different deployment scenarios, giving a strong foundation for analyzing market behavior.
3. Theoretical results are supported by numerical simulations, which help explain and deepen the insights from the model.
4. The findings offer practical recommendations for regulators and stakeholders on how open-source involvement can shape innovation outcomes.

> Weaknesses of the Paper:

1. The model oversimplifies the motivations of the open-source community by focusing only on performance and innovation, ignoring factors like ethics and societal goals.
2. The impact of hardware costs on deployment decisions is mentioned but not explored in enough detail.
3. The analysis is entirely theoretical and lacks validation through real-world examples or data.

**Questions:**

> Questions for the Authors:

1. How does the model handle situations where deployers use both open-source and closed-source solutions together (hybrid strategies)?
2. How would the results of the model change if real-world factors like regulations or data privacy rules were included?
3. Can you share examples or case studies showing how proactive open-source engagement has either helped or hurt innovation in practice?

**Reviewer Confidence:**

3: The reviewer is confident but not certain that the evaluation is correct

**Scope:**

4: The work is relevant to the Web and to the track, and is of broad interest to the community